# An effect of serotonergic stimulation on learning rates for rewards apparent after long intertrial intervals

Kiyohito Iigaya [1,2,3], Madalena S. Fonseca[4], Masayoshi Murakami[4], Zachary F. Mainen[4] & Peter Dayan [1,2]

Serotonin has widespread, but computationally obscure, modulatory effects on learning and cognition. Here, we studied the impact of optogenetic stimulation of dorsal raphe serotonin neurons in mice performing a non-stationary, reward-driven decision-making task. Animals showed two distinct choice strategies. Choices after short inter-trial-intervals (ITIs) depended only on the last trial outcome and followed a win-stay-lose-switch pattern. In contrast, choices after long ITIs reflected outcome history over multiple trials, as described by reinforcement learning models. We found that optogenetic stimulation during a trial significantly boosted the rate of learning that occurred due to the outcome of that trial, but these effects were only exhibited on choices after long ITIs. This suggests that serotonin neurons modulate reinforcement learning rates, and that this influence is masked by alternate, unaffected, decision mechanisms. These results provide insight into the role of serotonin in treating psychiatric disorders, particularly its modulation of neural plasticity and learning.

[1] Gatsby Computational Neuroscience Unit, University College London, 25 Howland Street, London W1T 4JG, UK. [2] Max Planck UCL Centre for Computational Psychiatry and Ageing Research, Russell Square House, 10-12 Russell Square, London WC1B 5EH, UK. [3] Division of Humanities and Social Sciences, California Institute of Technology, 1200 E California Blvd, Pasadena, CA 91125, USA. [4] Champalimaud Research, Champalimaud Centre for the Unknown, Avenida Brasília, 1400-038 Lisbon, Portugal. Correspondence and requests for materials should be addressed to K.I. (email: kiigaya@gatsby.ucl.ac.uk)

Learning from the outcomes of past actions is crucial for effective decision-making and thus ultimately for survival. In the case of important outcomes, such as rewards, ascending neuromodulatory systems have been implicated in aspects of this learning due to their pervasive effects on processing and plasticity. Of these systems, perhaps best understood is the involvement of phasically fluctuating levels of dopamine activity and release in signaling the so-called temporal difference[1] prediction errors (mismatches between outcomes and predictions) for appetitive outcomes[2,3]. Since prediction errors are a key component of reinforcement learning (RL) algorithms, this research has underpinned and inspired a large body of theory on the neural implementation of RL.

From the early days of investigations into aversive processing in Aplysia[4], serotonin (5-HT) has also been implicated in plasticity. This is broadly evident in the mammalian brain, from the restoration of the critical period for the visual system of rodents occasioned by local infusion of 5-HT[5] to the impairment of particular aspects of associative learning arising from 5-HT depletion in monkeys[6,7]. Despite theoretical suggestions for an association with aversive learning[8–12], direct experimental tests into serotonin's role in RL tasks have led to a complex pattern of results[13–18]. For instance, recent optogenetic studies reporting that stimulating 5-HT neurons could lead to positive reinforcement[15] do not appear to be consistent with other optogenetic findings, which instead suggest an involvement in patience[16,17,19] and even locomotion[18].

Here, we study a different aspect of the involvement of 5-HT in RL. Although prediction errors are necessary signals for learning, they are not sufficient. This is because there is flexibility in setting the learning rate, i.e., the amount by which an agent should update a prediction based on such errors. The learning rate determines the timescale (e.g., how many trials) over which reward histories are integrated to assess the value of taken actions, and it is naturally associated with neural plasticity (e.g., refs. [20–23]). 5-HT can readily influence learning rates through its interaction with dopamine[24]; and indeed, there is evidence that animals adapt the timescales of plasticity to the prevailing circumstances[23,25–28], and also consider more than one timescale simultaneously[29–32]. Thus, 5-HT could be involved in some, but not other, timescales for learning. It could also be associated with some, but not other, of the many decision-making systems[33–36] that are known to be involved in RL.

We therefore reanalyzed experiments in which mice performed a partially self-paced, dynamic foraging task for water rewards[17]. In this task, 5-HT neurons in the dorsal raphe nucleus (DRN) were optogenetically activated during reward delivery in a trial-selective manner. The precise control of the timing and location of stimulation offered the potential of studying in detail the way in which 5-HT affects reward valuation and choice. We used methods of computational model comparison to examine these various possible influences. We first noted a substantial difference in the control of actions that followed short and long intertrial intervals: only the latter was influenced by extended reward histories, as expected for choices driven by conventional RL. We then found that the learning rate associated with these (latter) choices was significantly increased by stimulations of DRN 5-HT neurons.

## Results

### Animals showed a wide distribution of inter-trial-intervals.
We reanalyzed data from a dynamic foraging or probabilistic choice task in which subjects faced a two-armed bandit[17]. Full experimental methods are given in the "Methods" section and in ref. [17]. Briefly, the subjects were four adult transgenic mice expressing

CRE recombinase under the serotonin transporter promoter (SERT-Cre[37]) and four wild-type littermates (WT)[17]. In this task (Fig. 1a), mice were required to poke the center port to initiate a trial. They were then free to choose between two side ports, where reward was delivered probabilistically at both ports on each trial (on a concurrent variable-ratio-with-hold schedule[38]). On a subset of trials, when mice entered a side port, 1 s of photo-stimulation was provided to DRN 5-HT neurons via an implanted optical fiber (Fig. 1b). ChR2-YFP expression was histologically confirmed to be localized to the DRN in SERT-Cre mice (Fig. 1c)[17].

Following previous experiments in macaque monkeys[29,38,39], the probability that a reward is associated with a side port per trial was fixed in a given block of trials (left vs right probabilities: 0.4 vs 0.1, or 0.1 vs 0.4). Once a reward had been associated with a side port, the reward remained available until collection (although multiple rewards did not accumulate). Photo-stimulation was always delivered at one of the side ports in a given block (left vs right probabilities: 1 vs 0, or 0 vs 1). Block changes occurred every 50–150 trials and were not signaled, meaning that animals needed to track the history of rewards in order to maximize rewards.

As previously reported[17], subject's choices tended to follow changes in reward contingencies (Fig. 1d), exhibiting a form of matching behavior[29,38,39]. A deterministic form of matching behavior can maximize average rewards in this task[32,40–42] because the probability of getting a reward increases on a side as the other side is exploited (due to the holding of rewards). For more behaviorally realizable policies, slow learning of reward contingencies has been shown to be beneficial to increase the chance of obtaining rewards[32].

We confirmed the results of previous analyses[17] showing that the optogenetic stimulation of DRN neurons did not appear to change the average preference of the side ports (Fig. 1e). The animals' preference for the side port that was associated with a higher water probability was not affected by the side which was photo-stimulated. However, these analyses do not fully take advantage of the experimental design in which photo-stimulation was delivered on a trial-by-trial basis. This should allow us to examine whether the effect of stimulation is more prominent on a specific subset of trials.

### Duration of preceding ITI determined decision strategy.
The task contained a free operant component in that the subjects were free to initiate each trial. This resulted in a wide distribution of inter-trial-intervals (ITIs). It was notable that some ITIs were substantially larger than others (Fig. 1f; see also Supplementary Figs. 1 and 2). To quantify this effect, we separated short from long ITI trials using a threshold of 7 s (we consider other thresholds below; values greater than 4 s led to equivalent results; we set this threshold as a free parameter in our computational model analysis).

Supplementary Figure 3 reports the mean proportions of long ITI trials in WT and SERT-Cre mice. The frequency of long ITI trials was slightly, but statistically significantly, different between WT and SERT; however, this appears not to be due to stimulation itself, as control analysis showed that stimulation itself did not significantly change the ITI that followed (Supplementary Fig. 4). We also found that long ITI trials were most common in the last part of each experimental session, but were also seen in earlier parts of each session (Supplementary Fig. 5).

Previous studies have suggested a relationship between the duration of an ITI and the nature of the subsequent choice. For example, subjects have been reported to make more impulsive choices after shorter ITI[43]. Another study has shown that perceptual decisions are more strongly influenced by more

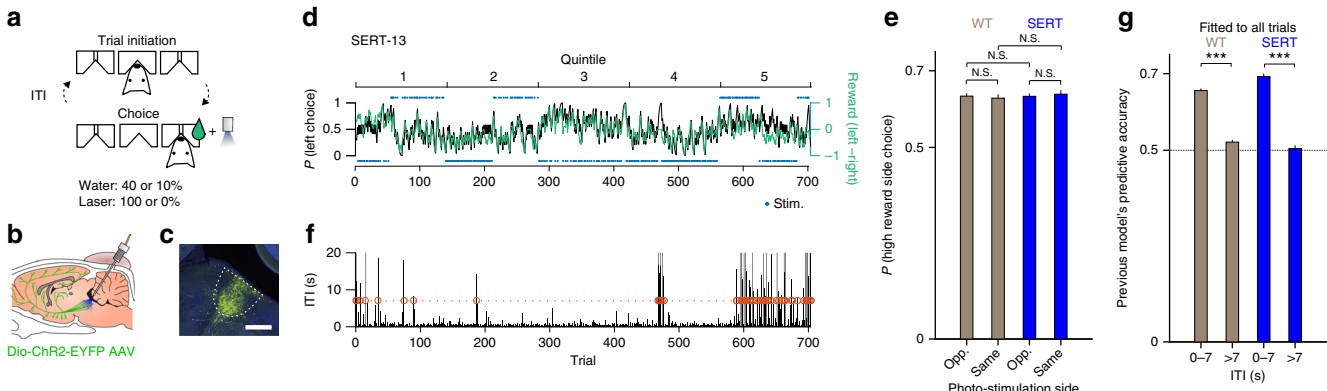

**Fig. 1** Task and behavior. **a** The task. On each trial, a mouse was required to enter the center port (Trial initiation) and then move to one of the side ports (Choice). A reward might be delivered at the side port according to a reward schedule. The next trial started when the mouse entered the center port. The inter-trial-interval (ITI) is defined as the time from when the mouse left the side port until it entered the center port to initiate the next trial. In a given block of trials, one side port was associated with a higher reward probability per trial (0.4) than the other (0.1), while photo-stimulation was always delivered at one of the side ports. **b** A schematic of the optogenetic stimulation. In SERT-Cre mice, 5-HT neurons expressed ChR2-YFP (green) and could be photoactivated with blue light. **c** A fluorescence image of a parasagittal section shows localized ChR2-YFP expression (YFP = green, DAPI = blue) in the DRN. The white bar indicates 500 μm. **d** Time course of mouse choice behavior in an example session. The probability across trials of choosing the left port (black solid line) is overlaid with the collected reward bias (green line) for an example mouse, SERT-13. The choice probability and the reward bias were computed by a causal half Gaussian filter with a standard deviation of two trials. The top (bottom) light blue dots indicate photo-stimulation at left (right) port. **e** The probability of choosing the higher water probability side for the blocks in which the photo-stimulation was assigned to the opposite side from the higher water probability side (Opp.), and for the blocks in which the photo-stimulation was assigned to the same side (Same). The difference within WT mice, within SERT-Cre mice, and between WT and SERT-Cre mice for either condition were not significant. The error bars indicate the mean ± SEM over sessions. **f** ITIs in the same session as **d**. The red circle indicates trials with long ITIs (>7 s). **g** The average predictive accuracy of the existing reward and choice kernel model[17,38] when fitted to all trials. This model captures a form of win-stay, lose-shift rule. Choices following short ITIs (≤7 s) were well predicted by the model, while choices following short ITIs (>7 s) were not. The difference between short and long ITIs was significant for both WT and SERT mice (permutation test. $p < 0.001$, indicated by three stars.) Images **b**, **c** are reproduced from ref. [17] (Copyright [2015], Elsevier)

venerable prior experience when working memory was disturbed during the task[44]. Here, we hypothesized that choices following short ITIs might also be more strongly influenced by the most recent choice outcome compared to those following long ITIs, since, for example, the outcome preceding a short ITI is more likely to be kept in working memory until the time of choice.

To investigate this, we first exploited an existing model of the behavior on this task[17,38]. This is a variant of an RL model which separately integrates reward and choice history over past trials, subject to exponential decay[38]. This model captures a form of win-stay, lose-shift rule[45,46] when time constants are small.

We found that choices following short ITIs (ITIs < 7 s) were well predicted by this previously validated model (see "Methods" for details) (Fig. 1g). Further, the time constants of the model were indeed very short (reward kernel: 1.4 trials for WT and 1.9 trials for SERT-Cre mice; choice kernel: 1.3 trials for WT and 1.2 trials for SERT-Cre mice). This suggests choices followed a form of win-stay, lose-shift rule[45,46]. The difference of the reward time constant between WT and SERT-Cre mice was significant ($p <$ 0.01, permutation test) but very small (<1 trial), while the choice time constant was not. This paltry difference in reward time constant suggests a slightly smaller learning rate for the SERT-Cre mice, since the learning rate is inversely proportional to the time constant.

However, choices following long ITIs (ITIs > 7 s) were not well predicted by the same model when fitting the model to all trials (Fig. 1g), suggesting that choices following short ITIs and long ITIs are qualitatively different. This is also evident from our additional parametric analysis showing that predictive accuracy of the win-stay lose-switch strategy dramatically decreased as ITIs lengthened (Supplementary Fig. 6). This did not depend on whether long ITI trials were in the beginning of, or in the last part of, each experimental session (Supplementary Fig. 7; being at, or

slightly below, chance). These results also suggest that choices following long ITIs cannot be accounted for by a short-term memory-based win-stay lose-switch strategy.

We hypothesized that choices following long ITIs might reflect slow learning of reward history over many trials[32,47]. We first fit the same kernel model only to choices following long ITIs, by allowing the model to learn over all trials but maximizing the likelihood only from the choices following long ITIs. We found that the model could now well predict choices following long ITIs, while failing to account for choices following short ITI (Supplementary Fig. 8). Further, the time constants of the model were now very long (reward kernel: 91 trials for WT and 59 trials for SERT-Cre mice; choice kernel: 100 trials for WT and 143 trials for SERT-Cre mice). This supports the idea that choice following long ITIs were driven by slow learning of outcomes over many trials. We should note, however, that the difference between the choice and reward kernels becomes somewhat obscure over this timescale, since the reward and choice histories are strongly correlated over the long run. Thus, one should take this result as inspiration, and be cautious about interpreting the precise parameter values.

We then looked for the best model that can describe choice following long ITIs. As noted above, the kernel model is likely to have a redundant structure for slow learning. Indeed, by complexity-adjusted model comparison (integrated BIC)[48,49], we found that choices following ITIs > 7 s were best described by a standard RL model (Supplementary Fig. 9). This analysis again supports our hypothesis that choices following long ITIs are influenced by a relatively long period of reward history compared to choices following short ITIs. It is also worth noting that in contrast to the short ITI model, in which memory decays rapidly every trial regardless of choice, the standard RL model does not change the value of an option as long as the option is not selected.

This difference in models could suggest that different memory mechanisms may be involved in the decisions following short and long ITIs (e.g., working memory for short ITIs, longer memory for long ITIs[35]).

**Enhanced learning from DRN stimulation**. Given our original hypothesis that serotonin modulates the RL learning rate, we predicted that optogenetic stimulation of DRN 5-HT neurons would have a stronger impact on future choices that follow long ITIs, since those choices appear to be more sensitive to learning over long trial sequences.

To test this, we first conducted the model-agnostic analysis described schematically in Fig. 2a. To assess how reward history with or without photo-stimulation affected choice following long ITIs, we estimated correlations between the temporal evolution of the reward bias and the choice bias. Importantly, we estimated the reward bias on trials preceded by ITIs of any length, but separately for trials with or without photo-stimulation, while the choice bias was estimated on trials preceded by long ITIs, regardless of the presence of reward or photo-stimulation.

As seen in Fig. 2b, we found significant correlations between reward and choice bias for all conditions. Importantly, there was a significant effect of photo-stimulation on the magnitude of the correlation. That is, for the SERT-Cre mice, the correlation was larger when reward bias was estimated from trials with stimulation than when it was estimated from trials without

stimulation. This suggests that optogenetic stimulation of DRN 5-HT neurons modulated learning about reward history (independent of the ITI on the learning trial), which in turn affected future choices on trials that followed long ITIs.

The equivalent analysis for choices following short ITIs (Supplementary Fig. 10) showed that they were not affected by photo-stimulation. Indeed, a direct comparison between choices following short and long ITI conditions shows that the stimulation had a larger impact on reward learning for choices following long ITIs than for choices following short ITIs in SERT-Cre mice, while there was no difference in WT mice (Supplementary Fig. 11).

In addition, in the absence of photo-stimulation during reward deliveries, the correlation was smaller for the SERT-Cre mice than the WT mice (Fig. 2b). This could indicate a chronic effect of stimulation[18], or a baseline effect of the genetic constructs, in addition to the trial-by-trial effect.

**Faster reinforcement learning from DRN stimulation**. Our analysis so far suggests that choices following short ITIs are captured by a relatively simple win-stay lose-shift rule, while choices following long ITIs reflect a more gradual learning about reward and choice histories over multiple trials. Furthermore, we showed that optogenetic stimulation of 5-HT neurons at reward deliveries influenced the impact of those rewards on future choices following long, but not short, ITIs.

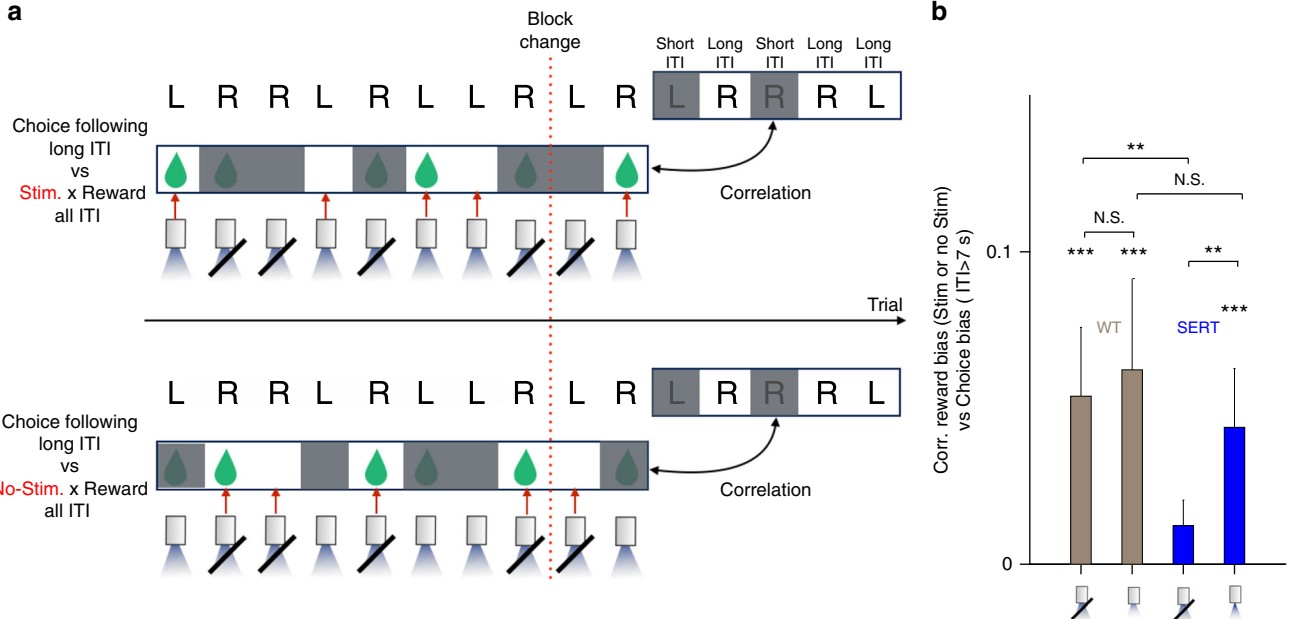

**Fig. 2** Enhanced learning from DRN stimulation. **a** Schematic diagram of the model-agnostic analysis. The correlation between the recent reward bias (window = 10 trials) and choices following long ITIs (window = 5 trials) was estimated using adjacent sliding windows. The reward bias was estimated on trials only with (top) or without (bottom) photo-stimulation, but regardless of the duration of ITIs. The choice bias was estimated only for choices following long ITIs, regardless of the presence of stimulation or reward. The grayed out trials in this example were ignored for the assessments. The windows were shifted together one trial at a time. For each realization of the sliding windows, the reward and adjacent choice biases were estimated. However, we excluded cases in which the choice window contained no long ITI trials. By sliding these windows, we obtained many pairs of reward bias and choice bias. We then estimated Pearson's correlation from these points, separately for each mouse. Note that, due to the task design in which photo-stimulation is associated with only one side (left or right) in a given block, in some moving windows reward bias had to be computed from one side only. Thus, we assigned +1 (respectively −1) to a reward from left (right) and no-reward from right (left) when we computed reward bias. We are aware that this is not a perfect measure for reward bias; but we still expect finite correlations since reward rates from the left choice and the right choice are on average negatively correlated by the task design in a given block (reward probability: 0.1 vs 0.4). **b** Model-agnostic analysis suggests that the impact of reward history on choices following long ITIs was modulated by optogenetic stimulation. The x-axis indicates if the reward bias was computed over trials with or without photo-stimulations. The stars indicate how significantly the correlation is different from zero, or the correlations are different from each other, tested by a permutation test, where estimated reward bias was permuted within or between conditions. Three stars indicates p < 0.001. The error bars indicate the mean ± SEM of subjects (n = 4)

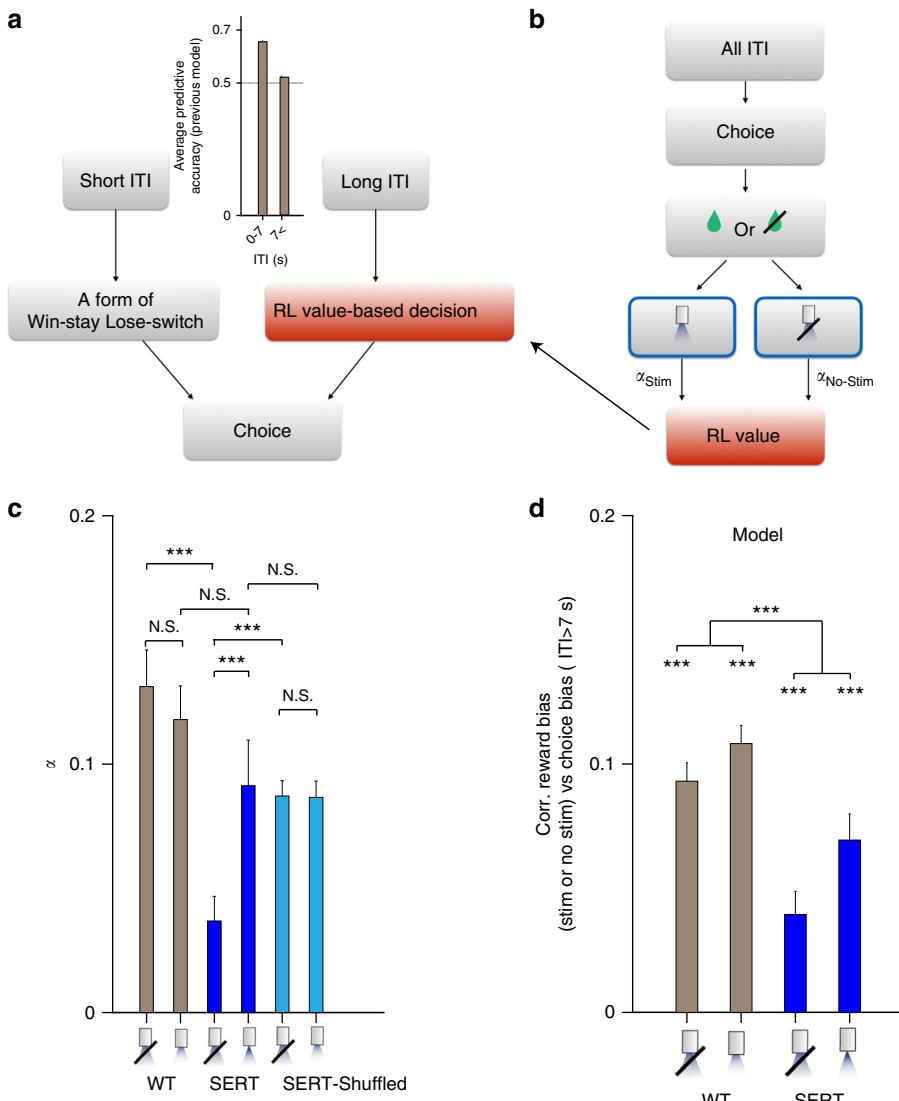

**Fig. 3** DRN 5-HT stimulation increased the learning rate of SERT-Cre mice. **a** Schematics of the computational model. There are two separate decision-making systems: a fast system generating a form of "win-stay, lose-switch," and a slow system following reinforcement learning (RL). After short ITIs ($T_{ITI} < T_{Threshold}$), choice is generated by the fast system following win-stay, lose-switch. After long ITIs ($T_{ITI} > T_{Threshold}$), choice is generated by the slow RL system. The ITI threshold $T_{Threshold}$ is a free parameter that is fitted to data. **b** The RL system is assumed to learn the value of choice on all trials, including those with short ITIs for whose choices it was not responsible. The learning rate of the RL system is allowed to be modulated by photo-stimulation. When photo-stimulation is (respectively, is not) delivered, choice value is updated at the rate of $\alpha_{Stim}$ ($\alpha_{no-Stim}$). The error bars indicate the mean ± SEM of sessions ($n = 32$). **c** Photo-stimulation increased the learning rate of SERT-Cre mice. The estimated learning rates for the WT (left), SERT-Cre (center), and SERT-Cre mice with shuffled stimulations (right) are shown. The difference between $\alpha_{Stim}$ and $\alpha_{no-Stim}$ in WT mice, between $\alpha_{Stim}$ in WT mice and $\alpha_{Stim}$ in SERT-Cre mice, between $\alpha_{Stim}$ and $\alpha_{no-Stim}$ in SERT-Cre mice with shuffled stimulation conditions, and between $\alpha_{Stim}$ in SERT-Cre mice and $\alpha_{Stim}$ in SERT-Cre mice with shuffled stimulation (right) conditions were not significant. The difference between $\alpha_{Stim}$ and $\alpha_{no-Stim}$ in SERT-Cre mice, and between $\alpha_{no-Stim}$ in WT and $\alpha_{no-Stim}$ in SERT-Cre mice were significant (permutation test, $p < 0.001$). The difference between $\alpha_{no-Stim}$ in SERT-Cre mice and $\alpha_{no-Stim}$ in SERT-Cre mice with shuffled stimulations was also significant (permutation test, $p < 0.01$). **d** Generative test of the model. The analysis of Fig. 2b was applied to data generated by the model. The correlations were all significantly different from zero, while the difference between photo-stimulation and no photo-stimulation conditions between WT and SERT-Cre mice was also significant

In order to understand these findings in a more integrated way, we built a combined characterization of choice. Figure 3a depicts a model in which there is an ITI threshold (now treated as a free parameter rather than being set to 7 s) arbitrating whether the previously validated two-kernel model[17,38] (i.e., short-term memory-based win-stay lose-switch model), or a longer-term reinforcement learning (RL) model[50] would determine choice. The RL model allowed for two different learning rates associated with the prediction error on a given trial (Fig. 3b): $\alpha_{Stim}$ (for stimulated trials) and $\alpha_{No-Stim}$ (for non-stimulated ones).

Importantly, both of the mechanisms learned values in parallel every trial; but choices were generated by one of the mechanisms according to the duration of the preceding ITI, where the ITI threshold was a free parameter that was fit to the data.

This model well predicts choices following short and long ITIs (Supplementary Figs. 12 and 13). We also found that this model fits the data more proficiently than a number of variants (see the "Methods" section for details) embodying a range of different potential effects of optogenetic stimulation. This includes acting as a direct reward itself; as a multiplicative boost to any real

reward; or causing a change in the learning and/or forgetting rates (Supplementary Fig. 14).

One might wonder if the behavior could be better accounted for by a model specifying forgetting as a function of elapsed time, including the ITIs. To test this, we constructed a model that learns and forgets outcome history according to wall-clock time (measured in seconds) rather than according to the number of trials. For this, we simply adapted the previously validated two-kernel model that integrates choice and reward history over trials[17,38] such that the influence of historical events is determined by how many seconds ago they happened, using the factual timing of the experiments. Our model comparison analysis using WT mice, however, substantially favored the account of trial-based model in Fig. 3a ($\Delta$ iBIC = 218). Introducing two time constants to the reward integration kernel did not change this conclusion.

In the best fitting model (Fig. 3a), we found that optogenetic stimulation increased the learning rate on stimulated trials in SERT-Cre mice, but not in WT mice (Fig. 3c). Consistent with the previous analyses, we also found that the time constants for the choice kernel and the reward kernel for choices following short ITIs were very short for both WT and SERT-Cre mice (Supplementary Fig. 15), and that the ITI thresholds were not significantly different between WT and SERT-Cre mice (Supplementary Fig. 16). In addition, we replicated the same results using a model with a fixed (=7 s) ITI threshold (Supplementary Fig. 17).

As a control analysis, we fitted the model to SERT-Cre data with randomly re-assigned stimulation trials. Shuffling the trials abolished the effect of photo-stimulation on the learning rate (Fig. 3c), supporting the hypothesis that the modulation of learning rates was caused by stimulation of DRN serotonin neurons.

Although the learning rate on stimulation trials in SERT-Cre mice was significantly greater than that on non-stimulated trials, it was not significantly different from the learning rate in WT mice (Fig. 3c), as already hinted by the model-agnostic analysis (Fig. 2b).

Finally, we performed a generative test of the model to assess its ability to capture key aspects of the data. To do this, we simulated our model 100 times using each collection of parameters fit to each session of each subject, and analyzed generated data using the model-agnostic procedures adopted for the original data (shown in Fig. 2b). We used the ITIs from the real data in determining which trial was preceded by a long or a short ITI when simulating choices from the model. The ITI threshold was given by the model. Consistent with the real data, the simulated data also showed a significant correlation between reward history and the choice after long ITIs, and a significant difference between photo-stimulation and no photo-stimulation conditions between WT and SERT-Cre mice (Fig. 3d).

Our analysis has so far focused on the impact of reward history over a relatively short timescale (<50 trials), compared to the length of a whole experimental session (>500 trials). Since animals can also learn reward histories over much longer timescales[29,32], and 5-HT neurons have shown to encode reward rates over multiple timescales[51], it is possible that the optogenetic stimulation of DRN neurons might have had effects over hundreds of trials. To examine this, we conducted a simple correlation analysis by dividing each session into five quintiles (containing equal numbers of trials) as in Fig. 1d and Supplementary Fig. 5, and asked how the choices following long ITIs in the last quintile (the only one with substantial numbers of long ITI choices) were correlated with the reward history stretched over all numbers of preceding quintiles (e.g., only the fifth, the fourth, the fifth, etc.). For reward history, we used the

probabilities determined by the experimenters rather than those observed by the subjects to avoid any bias that is independent of the reward history (such as choice history).

Choices following long ITIs were indeed significantly influenced by long run reward history spanning over the entire experimental session (Supplementary Fig. 18). The data from the generative test also confirm this correlation (Supplementary Fig. 18), albeit to a lesser degree, perhaps because the model only involves a single time constant and may thus have an inflated learning (and thus forgetting) rate relative to these long gaps. Furthermore, although the data show that these effects were stronger in SERT-Cre mice than in WT mice (two-way ANOVA; $p = 0.0016$, $p = 11.98$), we did not see this in our generative test results. Thus, the longer-time constants (slower learning) that are present[23,29,32] may also be affected by genotype or actual optogenetic stimulation.

## Discussion

There have been many suggestions for the roles that serotonin might play in decision-making and choice. These include ideas about influences over motor behavior[52], punishment[8,53,54], opponency with dopamine[10,11,24], satiation[55], discounting[56], patience[16,17,19], and even aspects of reward[15,51,57]. Here, we report an additional effect: stimulations of DRN 5-HT neurons can increase the rate at which animals learn from choice outcomes in dynamic environments.

A standard learning rule in RL has two distinct components. The first is the reward prediction error (RPE), which quantifies the difference between the actual and predicted value of outcomes. The phasic activity[2] of midbrain dopamine neurons and the local concentration of dopamine[58,59] in target regions has shown to follow this pattern. The second component is the learning rate, which determines how much change is actually engendered by the prediction error. From a normative perspective, learning rates are determined by the degree of uncertainty[25] —influenced by factors such as initial ignorance and the volatility of the environment, since we should only learn when there is something that we do not know. There is experimental evidence that this is indeed the case in animals and humans[26,27,60,61]. While it has been suggested that the neuromodulators norepinephrine (NE) and acetylcholine (ACh) may influence learning rates[28,62], our findings suggest that 5-HT DRN neurons also play a critical role. The interaction between 5-HT and dopamine could potentially be implicated in this effect, as various serotonin receptor types can increase the release of dopamine[63], which, if operating at an appropriate timescale, could boost the effective learning rate.

It is notable that the effect of altered learning rates was only apparent in choices on trials following long, but not short, ITIs (though it is critical to remember that stimulation affected learning on all stimulated trials regardless of ITIs, but that the effect of that learning could only be observed on future choices immediately following long ITIs). The former choices also hewed to a different strategy than the latter. Short ITIs appeared to lead to decisions closer to win-stay, lose-shift, meaning that subjects weighed barely more than the outcome of the most recent trial in their decision. The shift between strategies might correspond to a difference between a policy based on working memory[35] for very recent events (a few seconds) vs a plasticity-based mechanism like that assumed by standard incremental RL for incorporating events over longer periods. Note, though, that the boundaries between working memory and RL are becoming somewhat blurred[64]. It has been suggested that memory-based methods contribute to model-based control, by contrast with incremental model-free RL[33,36,64,65]; but this remains to be pinned down

experimentally. Note that a similar effect has been also observed in perceptual decision-making. In one example, longer-lasting prior experience was more influential when working memory was disturbed during the task[44].

The distribution of short and long ITI trials suggests that they might reflect the animal's motivational state as being high and low, respectively. Long ITI choices were most frequent in the last quintile of each experimental session, where animals were likely to be sated. That they also occurred in the beginning of experimental sessions might suggest that the subjects were not fully engaged in the task at the start, perhaps hoping to get out from the experimental chamber. A more systematic analysis of behavior during long ITIs would be required to uncover the nature of those events.

The fact that only a subset of trials was apparently affected by the stimulation is arguably a cautionary tale for the interpretation of optogenetics experiments. What looked like a null effect[17] had to be elucidated through computational modeling. Equally, for the short ITI trials, what seemed like behavior controlled by conventional RL, might come from a different computational strategy (and potentially neural substrate) altogether[35]. This could prompt a reexamination of previous data (as shown in ref. [64]). Further caution might be prompted by the observation that the learning rate in the SERT-Cre mice in the absence of stimulation was actually significantly lower than that of the WT mice in the absence of stimulation, rising to a similar magnitude as the WTs, with stimulation. This may be due to chronic effects of optogenetic stimulation of DRN neurons, as suggested in recent experiments[18]. For example, SERT-Cre mice may have been less motivated. Contrary to this, there was no difference in reward rates (Supplementary Fig. 21), and the stimulation itself did not change the duration of the subsequent ITI (Supplementary Fig. 4). Another possibility for this is due to baseline effects of the genetic constructs.

The learning rates that we found even for the slow system are a little too fast to capture fully the long-term correlation that can be found in the data. This is apparent in our additional analysis showing the correlation between the reward bias in the first quintile of the trials and the choice bias in the fifth quintile of the same session (Supplementary Fig. 19), also the correlation between the reward bias in the fifth quintile of the trials and the choice bias in the first quintile in the succeeding experimental session (Supplementary Fig. 20). The former is surprising, since it spans a large number of trials; the latter because it usually spans more than a day. This could suggest that learning in fact took place over a wide range of timescales, and the time constant that we found by our model fitting reflects a weighted average of those multiple time constants[23,32]. It would be interesting to study how the duration of ITI, or the level of engagement in the task, can change the weight or relative contribution of those distinctive time constants. It is plausible that the two decision strategies that we considered here are just an approximation to a wider collection of strategies that operate over a wider range of distinctive timescales. It would then be interesting to ask why stimulations of DRN 5-HT neurons preferentially affected slower components. Further questions include whether 5-HT neuron's effects would be better captured as an influence on the relative weighting of different timescales[32] rather than the changes in time constants themselves that we assumed in the model fitting.

Though as a first approximation, we assumed that a hard threshold separates ITIs for taking one choice strategy from another on following trials, we expect that this can be improved upon. For instance, which rule determines choice is presumably controlled by other variables associated with the subjects' internal states, to which we had limited access in our current study. It is also possible that both decision strategies co-exist on every trial,

but their relative contributions to each ultimate decision are determined by some rules, as suggested for the integration of so-called model-based and model-free RL strategies[36]. In fact, there is evidence in macaque experiments that subjects shift to performing win-stay lose-switch if this cheap strategy offers a reasonable rate of rewards[66]. This is consistent with our finding that mice largely relied on the win-stay lose-switch, since switching behavior is known to be beneficial for this task (in fact, experimentalists often need to introduce a penalty (often in the form of a change-over delay) to deter such switching behavior[39,67]). Future studies in which the benefits and the costs of various strategies are manipulated would address this issue.

Finally, one of the main reasons to be interested in serotonin is the prominent role that drugs affecting this neuromodulator (such as selective serotonin reuptake inhibitors: SSRIs) play in treating psychiatric disorders. Our results suggest that serotonin boosts plasticity by influencing a form of learning rate. This resonates, for instance, with the fact that treatment with an SSRI can be more effective when combined with the so-called cognitive behavioral therapy (CBT), which encourages re-learning of habits in patients[68]. Our result that optogenetic stimulation of 5-HT neurons primarily impacted the slow RL system is consistent with this, under the assumption that this system is likely to be involved in habit formation on a longer timescale[33]. Thus, our results can offer the prospect of richer and more finely targeted manipulations.

## Methods

**The task**. Here, we briefly summarize experimental methods that have been described fully in ref. [17]. Eight mice (four SERT-Cre[37] and four wild-type littermates) were used for the probabilistic choice task.

At the beginning of each trial, an LED at the center port was illuminated to signal the mouse to insert its snout into the center port. Once the mouse entered the center port, the center LED extinguished and LEDs on choice ports located next to the center port were illuminated. The mouse could then choose one of the choice ports. Each choice port was associated with a specific reward probability (40 or 10%) and stimulation probability (100 or 0%) for the entire duration of a block of trials, which lasts 50–150 trials. The center port was illuminated 1.1 s after entering a choice port, so that the mouse can initiate the next trial.

The reward schedule followed previous experiments[38,39], where a reward was kept associated with a port until the mouse selects the port. Hernstein's matching behavior has been shown to approximate choice behaviors in this reward schedule[32,38,39,41].

Reward probabilities can never be the same for the two choice ports in a given block (if one side is high, the other is low). This is also the case for photostimulation. However, there was a slight bias in the block types: 25% of the blocks were associated with left high water probability and left photo-stimulation, 37.5% of the blocks were associated with left high water probability and right photo-stimulation, 28.1% of the blocks were associated with right high water probability and left photo-stimulation, and 9.4% of the blocks were associated with right high water probability and right photo-stimulation. The conclusion of our analysis was not affected by this bias.

If a water reward was assigned to the side chosen by the mouse, a drop of 3 μl of water was delivered at the port. If a photo-stimulation was assigned to the chosen side, a train of 10 ms, 5 mW, pulses was delivered for 1 s at 12.5 Hz.

Data were collected for 15 sessions (1 h and 15 min for each session) from all mice. Experimenters were blind to the mice genotype throughout the experiment, until histological analysis was performed afterwards[17]. All procedures were carried out in accordance with the European Union Directive 86/609/EEC and approved by Direcção-Geral de Veterinária of Portugal.

**Inter-trial-interval (ITI)**. We defined the inter-trial-interval (ITI) as the time from when the mouse left one of the side ports until it entered the center port to initiate the next trial. Occasionally, animals re-visited the side port long after their first visit on the same trial (<5% of all trials). These redundant pokes were ignored.

**Reward and choice kernel model**. Previous studies have shown that animals' choice behavior in a dynamic foraging task without the change-over-delay constraint[69] can be well described by a linear two-kernel model (e.g., refs. [17,38]). In this model, the probability $P_t^L$ of choosing left on trial $t$ is determined by a linear

combination of values computed from reward and choice history, given by

$$P_t^L = \frac{1}{1 + e^{-\left(a_t^L - a_t^R + b_t^L - b_t^R + \delta\right)}}, \tag{1}$$

where $a_t^L$ ($a_t^R$) is the value computed from a reward kernel for left (right), $b_t^L$ ($b_t^R$) is the value computed from a choice kernel for left (right), and $\delta$ is the bias. Assuming simple exponential kernels[29,38,39], the reward values are updated on every trial as:

$$a_{t+1}^L = (1 - \chi)a_t^L + \rho r^L \tag{2}$$

$$a_{t+1}^R = (1 - \chi)a_t^R + \rho r^R \tag{3}$$

where $a_t^L$ ($a_t^R$) is the reward value for left (right) choice on trial $t$, $\chi$ is the temporal forgetting rate of the kernel, $\rho$ is the initial height of the kernel, and $r^L = 1$ ($r^R = 1$) if a reward is obtained from left (right) on trial $t$, or 0 otherwise. Since these equations can also be written as:

$$a_{t+1}^L = a_t^L + \chi\left(\frac{\rho}{\chi}r^L - a_t^L\right) \tag{4}$$

$$a_{t+1}^R = a_t^R + \chi\left(\frac{\rho}{\chi}r^R - a_t^R\right) \tag{5}$$

this kernel is equivalent to a forgetful Q-learning rule[35,70] with a learning rate $\chi$ and reward sensitivity $\rho/\chi$.

The value for choice is also updated as:

$$b_{t+1}^L = (1 - \nu)b_t^L + \eta C^L \tag{6}$$

$$b_{t+1}^R = (1 - \nu)b_t^R + \eta C^R \tag{7}$$

where $b_t^L$ ($b_t^R$) is the choice value for Left (Right) choice on trial $t$, $\nu$ is the temporal forgetting rate of the kernel, $\eta$ is the initial height of the kernel, and $C^L = 1$ ($C^R = 1$) if Left (Right) is chosen on trial $t$ while 0 otherwise. We note that the initial height of the choice kernel, $\eta$, is normally negative[17,38], meaning that the choice kernel normally captures a tendency toward alternation. Such tendencies are common in tasks with reward schedules like those in the current task if a penalty for alternation is not imposed (change-over delay)[69].

We assumed that the update takes place on every trial, even those associated with long ITIs.

**Main model**. We constructed a model that describes choices on all trials. Since we found that the characteristics of decision strategies changed according to the ITIs, we simply assumed a two-agent model, where agent 1 (fast system) makes decisions on the trials following short ITIs (ITI $\leq T_{\text{Threshold}}$), while agent 2 (slow system) makes decisions on the trials following long ITIs (ITI $> T_{\text{Threshold}}$). We allowed the threshold $T_{\text{Threshold}}$ to be a free parameter that is determined by data. We also tested the fixed value $T_{\text{Threshold}} = 7$ s based on our preliminary analyses and found results consistent with the variable ITI threshold model (Supplementary Fig. 17).

The fast system generates decisions based on the two-kernel model described in "Reward and choice kernel model" section. The slow system performs simple Q-learning. Specifically, the probability $P_t^L$ of choosing Left on trial $t$ after a long ITI $> T_{\text{Threshold}}$ is given by

$$P_t^L = \frac{1}{1 + e^{-\left(v_t^L - v_t^R + \kappa\right)/T}}, \tag{8}$$

where $v_t^L$ ($v_t^R$) is the value for Left (Right), $\kappa$ is the bias term, and $T$ is the decision noise.

The agent updates values for chosen action according to the Rescorla–Wagner rule, but at different learning rates for photo-stimulation ($\alpha_{\text{Stim}}$) and no stimulation ($\alpha_{\text{No-Stim}}$) trials. For example, if Left was chosen and photo-stimulation was applied, the value of Left choice is updated as:

$$v_{t+1}^L = v_t^L + \alpha_{\text{Stim}}\left(r^L - v_t^L\right). \tag{9}$$

If no stimulation was applied, on the other hand,

$$v_{t+1}^L = v_t^L + \alpha_{\text{No-Stim}}\left(r^L - v_t^L\right). \tag{10}$$

By comparison with Eq. (4), we can see this as a non-forgetful Q-learner, but with a slightly more convenient parameterization for the reward sensitivity. For a model comparison purpose, we also fitted a forgetful Q-learner model with optogenetically modulated learning rates, in which the updates given by Eqs. (9) and (10) take place for the values of both choices every trial.

Both systems update values for every trial regardless of the preceding ITIs, but the decision was made by one of them depending on the most recent ITI, where the threshold $T_{\text{Threshold}}$ was also a free parameter. Figure 3 shows the results of this full model.

**Other models**. In order to explore other possibilities for optogenetic stimulation effects, we constructed three other models.

Asymmetric learning rate model: We allowed the model to have different learning rates for reward and no-reward trials when photo-stimulation was applied. Specifically, we modified Eq. (9) of the main model as:

$$v_{t+1}^L = v_t^L + \alpha_{\text{Stim}}^+\left(r^L - v_t^L\right) \tag{11}$$

if $r^L = 1$, and

$$v_{t+1}^L = v_t^L + \alpha_{\text{Stim}}^-\left(r^L - v_t^L\right) \tag{12}$$

if $r^L = 0$. The same is applied for the Right choice.

Multiplicative value model: Here we assumed that photo-stimulation changed the sensitivity of reward. Specifically, we modified the learning rules of slow system as:

$$v_{t+1}^L = v_t^L + \alpha\left(G_{\text{Stim}} \times r^L - v_t^L\right). \tag{13}$$

if photo-stimulation is applied, otherwise

$$v_{t+1}^L = v_t^L + \alpha\left(r^L - v_t^L\right). \tag{14}$$

Additive value model: Here we assumed that photo-stimulation carried a independent rewarding value. Specifically, we modified the learning rules of slow system as:

$$v_{t+1}^L = v_t^L + \alpha\left(G_{\text{Stim}} + r^L - v_t^L\right), \tag{15}$$

if photo-stimulation is applied, otherwise

$$v_{t+1}^L = v_t^L + \alpha\left(r^L - v_t^L\right). \tag{16}$$

The same is applied for the Right choice.

**Model fitting**. In order to determine the distribution of model parameters **h**, we conducted a hierarchical Bayesian, random effects analysis[48,49,71] for each subject. In this, the (suitably transformed) parameters $h_i$ of experimental session $i$ are treated as a random sample from a Gaussian distribution with means and variance $\theta = \{\mu_\theta, \Sigma_\theta\}$.

The prior distribution $\theta$ can be set as the maximum likelihood estimate:

$$\theta^{ML} \approx \text{argmax}_\theta\{p(D|\theta)\}$$
$$= \text{argmax}_\theta\left\{\prod_{i=1}^{N} \int d\mathbf{h_i}\, p(D_i|\mathbf{h_i})p(\mathbf{h_i}|\theta)\right\} \tag{17}$$

We optimized $\theta$ using an approximate Expectation–Maximization procedure. For the E-step of the k-th iteration, a Laplace approximation gives us

$$\mathbf{m}_i^k \approx \text{argmax}_h\left\{p(D_i|\mathbf{h})p(\mathbf{h}|\theta^{k-1})\right\} \tag{18}$$

$$p(\mathbf{h}_i^k|D_i) \approx \mathcal{N}\left(\mathbf{m}_i^k, \Sigma_i^k\right), \tag{19}$$

where $\mathcal{N}\left(\mathbf{m}_i^k, \Sigma_i^k\right)$ is the normal distribution with the mean $\mathbf{m}_i^k$ and the covariance $\Sigma_i^k$ that is obtained from the inverse Hessian around $\mathbf{m}_i^k$. For the M step:

$$\mu_\theta^{k+1} = \frac{1}{N}\sum_{i=1}^{N}\mathbf{m}_i^k \tag{20}$$

$$\Sigma_\theta^{k+1} = \frac{1}{N}\sum_{i=1}^{N}\left(\mathbf{m}_i^k\mathbf{m}_i^{k\mathbf{T}} + \Sigma_i^k\right) - \mu_\theta^{k+1}\mu_\theta^{k+1\mathbf{T}}. \tag{21}$$

For simplicity, we assumed that the covariance $\Sigma_\theta^k$ had zero off-diagonal terms, assuming that the effects were independent.

**Model comparison**. We compared models according to their integrated Bayes Information Criterion (iBIC) scores[48,49,71]. We analyzed model log likelihood log $p(D|M)$:

$$\log p(D|M) = \int d\theta\, p(D|\theta)p(\theta|M) \tag{22}$$

$$\approx -\frac{1}{2}\text{iBIC} = \log p\left(D|\theta^{ML}\right) - \frac{1}{2}|M|\log|D|, \tag{23}$$

where iBIC is the integrated Baysian Information Criterion, $|M|$ is the number of fitted prior parameters, and $|D|$ is the number of data points (total number of

choice made by all subjects). Here, $\log p(D|\theta^{ML})$ can be computed by integrating out individual parameters:

$$\log p(D|\theta^{ML}) = \sum_i \log \int d\mathbf{h}\, p(D_i|\mathbf{h}) p(\mathbf{h}|\theta^{ML}) \qquad (24)$$

$$\approx \sum_i \log \frac{1}{K} \sum_{j=1}^{K} p(D_i|\mathbf{h}^j), \qquad (25)$$

where we approximated the integral as the average over $K$ samples of $h^j$'s generated from the prior $p(\mathbf{h}|\theta^{ML})$.

**Model's average predictive accuracy**. We defined the model's average predictive accuracy as the arithmetic mean of the likelihood per trial, using each session's MAP parameter estimate. That is,

$$p(D_i|\mathbf{h}_i^{MAP}) = \frac{\sum_{t=1}^{N_{trial}} p(d_i^t|\mathbf{h}_i^{MAP})}{N_{trial}}, \qquad (26)$$

where $N_{trial}$ is the number of the trial, $d_i^t$ is the datapoint on trial $t$ in session $i$.

In our generative simulations, we used the same reward/photo-stimulation schedule as the actual data.

**Data availability**. The data and codes that support the findings of this study are available from the corresponding author on reasonable request.

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

## Acknowledgements

We thank the Gatsby Charitable Foundation, the Joint Initiative on Computational Psychiatry and Ageing Research between Max Planck Society and UCL, the Japan Society for the Promotion of Science, the European Research Council (250334 and 671251), Fundação para a Ciência e a Tecnologia (PD/BD/52446/2013 and SFRH/BPD/46314/2008), and the Champalimaud Foundation for generous support.

## Author contributions

K.I., M.S.F., M.M., Z.F.M., and P.D. conceived the project. M.S.F., M.M., and Z.F.M. designed and performed the original experiments. K.I. and P.D. developed computational models, designed and performed analysis with inputs from M.S.F., M.M., and Z.F.M. K.I., M.S.F., M.M., Z.F.M., and P.D. discussed the results and wrote the manuscript.

## Additional information

**Competing interests:** The authors declare no competing interests.

