## [Peer Review File · Nature Communications]

Reviewers' comments:

Reviewer #1 (Remarks to the Author):

Igaya and colleagues analyse a previously published dataset obtained from mice performing a foraging task. While mice selected between two options for probabilistically delivered water reinforcement, the authors optogenetically activated 5-HT neurons in the dorsal raphe nucleus on subsets of trials. The authors analysed the choice data subsetted by intertrial intervals, and claim that the learning rate associated with rewards delivered following long intertrial intervals was specifically increased by 5-HT stimulation. The subject is certainly interesting, the claims novel, and the careful analyses shed light on the puzzling result that 5-HT activation showed no appetitive or aversive effects previously reported by some of the authors. Indeed, the careful analyses highlight the power and complexity of careful computational analyses of behaviour, and I believe that the results could significantly influence thinking in the field. However, I find the analyses not yet sufficient to strongly support the claims, and have some comments/suggestions for further analyses that could clarify key issues.

Although the authors show that 5-HT activation increases the learning rate on some trials in their model, it seems more correct to say that SERT-cre mice have a deficit exposed by the task, which is normalised by photostimulation. This has hallmarks of a binary effect; SERT-cre mice show almost no learning from long ITI trials (Figure 2b), and stimulation brings the learning rate on these trials to the same level as WT mice (Figure 3c), but does nothing on trials with shorter ITIs (Figure S10). This could be interpreted (and possibly tested, see below) as if SERT-cre mice are off doing something else on long ITI trials, and photostimulation brings them back "on-task". This would be consistent with no-effect on learning on short ITI trials since the animals are already on-task. This kind of effect would masquerade as a change in learning rate.

It would be nice to see the actual ITI distributions for individual mice. While the authors state that 4 seconds leads to equivalent results, these cutoffs are a little hard to justify from the single session of data in Figure 1. Can the authors say something about what the mice were doing during the longer ITIs? The tails are potentially very interesting here. What is happening for 20+ second ITIs? In particular, is this different between WT mice and SERT-cre mice? Did WT and SERT-cre mice work for the same amounts of fluid/# of trials? This could give some insight into the suggestion that long ITIs are related to low motivational state.

It seems that a lot of useful information could be gleaned from no-stimulation sessions in SERT-cre mice. For example, it might indicate whether chronic stimulation leads to low learning on long ITI trials? Or whether chronic stimulation changes the distribution of ITIs? If this data is available, it would be useful to include it in the paper.

It would be useful to present predictive accuracy on >7 second ITI trials for the model in Figure 3. The low accuracy of the WSLS model on these trials is striking in Figure 1, and it would be nice to see what ground was gained in the two-component model.

I failed to understand the model-agnostic correlation analysis (Figure 2). It purports to show a correlation between two segments of data with different lengths (10 and 5 trials for rewards and choice respectively). Was this data somehow reduced to a single number for each step? The gaussian filter from Figure 1d? Please add just a bit more detail so readers can follow from Figure 2a to a pair of numbers that enter into vectors that are ultimately correlated. What kind of correlation?

I am definitely appreciate the generative model testing (Figure 3d). However, I'm a little confused in

this case since there is no model of ITI generation. How did the authors run the two-component model in this case? Are there simply no short ITIs in the generated data?

MINOR

Figure S10, the label says "all trials", but I think it should say "<7 second ITI".

Figure 1g, S1, the label "7<" might be easier for most to understand if it was changed to ">7"

Why do the authors use "partially" self-paced? It seems fully self-paced?

Reviewer #2 (Remarks to the Author):

Summary

In this manuscript the authors examine the effect of stimulating serotonin neurons on learning a self-paced probabilistic stimulus-response choice task. To do this the authors reanalyze existing data from a prior publication (Fonseca et al. Current Biol 2015) in which they examined the effect of ChR2-dependent stimulation of dorsal raphe serotonin neurons (using the Slc6a4::Cre line) on reward and waiting parameters. In this study they focus on the effect of serotonin neuron stimulation on response bias that depends on past response history. They use an error-prediction learning rate model to fit the data and estimate what type of decision strategy the animals use at each response choice. They make two major discoveries. First, they find that the animals use recent response history to guide choices when the inter-trial interval (ITI) is short (<7 s, under most circumstances) – a type of win-stay strategy, but use average response history following long ITIs (that occur sporadically). Second, they find that when they include serotonin neuron stimulation in their model it best fits the data when it influences learning rate for long ITIs. Based on these modeling data they argue that serotonin increases learning rates for reward. Understanding the role of serotonin neuromodulation in behavior is a major goal of neuroscience research with high clinical relevance. The data appear sound and are constructively imbedded in a logical experimental hypothesis and computational model and thus are an important addition to the field.

Comments:

1. The Slc6a4::Cre line they use is presumably a null allele of Slc6a4 and thus WT animals are not really the right control for these experiments. Several of the baseline, non-stimulation parameters appear to be affected by this genotype difference and the fact that the Slc6a4 mutation is likely to directly affect serotonin homeostasis, this confound is potentially problematic. The authors are aware of this point and are careful to point out where genotype may have influenced the data. However, they need to explicitly discuss the potential impact that heterozygosity of Slc6a4 could have on their findings. These animals are known to have altered serotonin tone and availability and have been widely studied as a model for the human low-expressing 5-HTT-LPR allele. Also, because there appears to be a significant difference in learning rate between WT and Slc6a4::Cre mice, there may be a ceiling effect in the WTs that confounds the data.
2. Although the authors are generally careful to refer to their manipulation as stimulation of serotonin neurons, on occasion they talk about stimulation of serotonin or about serotonin affecting learning rates. Given that serotonin neurons also release other neurotransmitters and these have been shown to be responsible for at least some of the phenotypes associated with stimulation of their cell bodies, the authors cannot infer that their effects are the result of changes in serotonin.

3. The title is overly baroque and misleading. The first phrase should be eliminated as it appears to be intended to play on the presence of the S and L alleles of the serotonin transporter, but the manuscript in fact does not refer to these.

4. Figure 1 is very small and hard to read. All the figures would benefit from being made easier for the eye.

5. The methods section relies on their earlier paper for many items. Better to reiterate the critical information here (e.g. animals, methods).

Reviewer #3 (Remarks to the Author):

Iigaya and colleagues analyzed data from a previously published experiment (Fonseca et al., 2015) in which dorsal raphe serotonin neurons were stimulated in mice performing a foraging task. They show that serotonin activation increased learning rates following long ITIs, concluding that serotonin changes learning rates in an RL context. The previously published experimental data are beautiful, the model is conceptually exciting, but I found the conclusions vastly overstated given the data, and model selection seemed arbitrary.

1. The boundary between long and short ITIs seems arbitrary. The claim on lines 132-133 is that "choices following short ITIs and long ITIs are qualitatively different." This ultimately leads to the conclusion that "different memory mechanisms may be involved in the decisions following short and long ITIs" (lines 143-144). Is there evidence for a nonlinearity in the effects as a function of ITI that would justify the arbitrary boundary? In any case, it would be useful to see a histogram of ITIs from one session and histograms of all ITIs for each mouse.

2. Are the effects driven by long ITIs at the end of sessions? In this case, the effects could be interpreted as "persistence" or "task engagement," rather than learning rates, per se. The authors show that most long ITIs were at the end of sessions (Fig. S3), but do not evaluate the contribution of time within a session to the reported effects.

Given that the ITIs were self-generated, it is difficult to disentangle forgetting (presumably a nondecreasing function of time) from "motivation" (a nonincreasing function of time, as the animal gets less thirsty). A new experiment, with experimenter-generated ITIs, could potentially resolve this.

3. Did the RL model with slow learning still fit the behavior following short ITIs? It seems very strong to conclude different memory mechanisms due to quantitatively different fits (BIC score differences). Indeed, a more parsimonious explanation would be simply that the new model is better at describing behavior than the old one (which, by itself, would be interesting). If the authors believe there are two separate memory mechanisms (issues above notwithstanding), how would it work for the brain to "choose" one over the other in real time? Is the claim that $T_{\text{Threshold}}$ is implemented neurally? Why would serotonin affect one but not the other? Why not parameterize ITI in the model, as opposed to using a threshold value?

Reviewers' comments:

Reviewer #1 (Remarks to the Author):

ligaya and colleagues analyse a previously published dataset obtained from mice performing a foraging task. While mice selected between two options for probabilistically delivered water reinforcement, the authors optogenetically activated 5-HT neurons in the dorsal raphe nucleus on subsets of trials. The authors analysed the choice data subsetted by intertrial intervals, and claim that the learning rate associated with rewards delivered following long intertrial intervals was specifically increased by 5-HT stimulation. The subject is certainly interesting, the claims novel, and the careful analyses shed light on the puzzling result that 5-HT activation showed no appetitive or aversive effects previously reported by some of the authors. Indeed, the careful analyses highlight the power and complexity of careful computational analyses of behaviour, and I believe that the results could significantly influence thinking in the field.

Thank you very much for the enthusiastic opinion about our manuscript.

However, I find the analyses not yet sufficient to strongly support the claims, and have some comments/suggestions for further analyses that could clarify key issues.

Although the authors show that 5-HT activation increases the learning rate on some trials in their model, it seems more correct to say that SERT-cre mice have a deficit exposed by the task, which is normalised by photostimulation. This has hallmarks of a binary effect; SERT-cre mice show almost no learning from long ITI trials (Figure 2b), and stimulation brings the learning rate on these trials to the same level as WT mice (Figure 3c), but does nothing on trials with shorter ITIs (Figure S10). This could be interpreted (and possibly tested, see below) as if SERT-cre mice are off doing something else on long ITI trials, and photostimulation brings them back "on-task". This would be consistent with no-effect on learning on short ITI trials since the animals are already on-task. This kind of effect would masquerade as a change in learning rate.

We appreciate this comment. But we first would like to clarify possible confusions here. There are two separate questions: which rules govern choice; and what mechanism or learning rates underlie those rules. We found that the rule governing choice differed according to whether the ITI was short or long; with the latter being determined by a learning algorithm. We also found that the learning rate on a trial was affected by stimulation. This stimulation effect on learning did not depend on whether the ITI was short or long on that trial; however, it was only after a subsequent long ITI trial that the choice that the animal made revealed this changed learning rate.

This is true for both our model-agnostic (Figure 2) and our model analysis (Figure 3). The model learns from all trials regardless of the duration of ITIs, but the learning rate is different according to the presence of stimulation. Choice, instead, is generated by one of the mechanisms determined by the most recent ITI.

We realize that this part must have been very confusing, and we apologize for this. This is now clarified in the Results and Discussions sections as:

(line 181): *To assess how reward history with or without photo-stimulation affected choice following long ITIs, we estimated correlations between the temporal evolution of the reward bias and the choice bias. Importantly, we estimated the reward bias on trials preceded ITIs of any length, but separately for trials with or without photo-stimulation, while the choice bias was estimated on trials preceded by long ITIs, regardless of the presence of reward or photo-stimulation.*

(line 188) Importantly, there was a significant effect of photo-stimulation on the magnitude of the correlation. That is, for the SERT-Cre mice, the correlation was larger when reward bias was estimated from trials with stimulation than when it was estimated from trials without stimulation. This suggests that optogenetic stimulation of DRN 5-HT neurons modulated learning about reward history (independent of the ITI on the learning trial), which in turn affected future choices on trials that followed long ITIs.

*The equivalent analysis for choices following short ITIs (**Figure S10**) showed that they were not affected by photo-stimulation in the same way. Indeed, a direct comparison between choices following short and long ITI conditions shows that the stimulation had a larger impact on reward learning for choices following long ITIs than for choices following short ITIs in SERT-Cre mice, while there was no difference in WT mice (**Figure S11**).*

(line 217) Importantly both of the mechanisms learned values in parallel every trial; but choices were generated by one of the mechanisms according to the duration of the preceding ITI, where the ITI threshold was a free parameter that was fit to the data.

(line 305) though it is critical to remember that stimulation affected learning on all stimulated trials regardless of ITIs, but that the effect of that learning could only be observed on future choices immediately following long ITIs.

Nonetheless, we were intrigued by the reviewer's idea that stimulation could make animals more engaged with the task. One way to test this is to see how long it took for animals to initiate the trials immediately after a stimulation (i.e. the duration of ITIs), assuming that animals would initiate the next trial particularly quickly if they were in a highly motivated state. We, however, did not find any effects of stimulation on subsequent ITIs (please see **Figure S4**). This is discussed in the manuscript:

*(line 335) This may be due to chronic effects of optogenetic stimulation of DRN neurons, as suggested in recent experiments. For example, SERT-Cre mice may have been less motivated. Contrary to this, there was no difference in reward rates (**Figure S21**), and the stimulation itself did not change the duration of the subsequent ITI (**Figure S4**).*

It would be nice to see the actual ITI distributions for individual mice. While the authors state that 4 seconds leads to equivalent results, these cutoffs are a little hard to justify from the single session of data in Figure 1. Can the authors say something about what the mice were doing during the longer ITIs? The tails are potentially very interesting here. What is happening for 20+ second ITIs? In particular, is this different between WT mice and SERT-cre mice? Did WT and SERT-cre mice work for the same amounts of fluid/# of trials? This could give some insight into the suggestion that long ITIs are related to low motivational state.

Following the reviewer's suggestion, we now include distributions of ITIs for all mice as new Figures S1, S2. We also plotted the average reward rate for both groups (please see new Figure S21), but found no significant difference between the groups. Further, we tested whether choices following long ITIs in the first parts of sessions and ones in the last parts of sessions were qualitatively different. We found no evidence for this (new Figure S7). This is discussed in the manuscript:

*(line 141) However, choices following long ITIs (ITIs > 7 s) were not well-predicted by the same model when fitting the model to all trials (**Figure 1g**), suggesting that choices following short ITIs and long ITIs are qualitatively different. This is also evident from our additional parametric analysis showing that predictive accuracy of the win-stay lose-switch strategy*

dramatically decreased as ITIs lengthened (**Figure S6**). This did not depend on whether long ITI trials were in the beginning of, or in the last part of, each experimental session (**Figure S7**; being at, or slightly below, chance). These results also suggest that choices following long ITIs cannot be accounted for by a short-term-memory-based win-stay lose-switch strategy.

Occasional video recordings of the task show that mice behaviors were very heterogeneous during long ITIs. This includes grooming, moving along the wall of chamber, and stopping at various positions. We agree that characterizing the nature of behaviors during ITIs, and determining what caused long ITI events, would be important steps toward a better understanding of behavior. For this, we would need to develop an automated algorithm to extract behavioral features dynamically from videos during ITIs. We plan to address this in our future projects.

It seems that a lot of useful information could be gleaned from no-stimulation sessions in SERT-cre mice. For example, it might indicate whether chronic stimulation leads to low learning on long ITI trials? Or whether chronic stimulation changes the distribution of ITIs? If this data is available, it would be useful to include it in the paper.

We agree with the reviewer, but unfortunately there was no no-stimulation session in the main task (one of the two choice ports was always associated with stimulation). Though stimulation was not given during training periods, the task was designed with a specific focus on stimulation. Thus, the pre-stimulation shaping schedule was tailored to the individual animals' performance, and so would be hard to generalize. Furthermore, the training sessions were shorter than the experimental ones. Thus we cannot perform reliable estimation for the training sessions.

It would be useful to present predictive accuracy on >7 second ITI trials for the model in Figure 3. The low accuracy of the WSLS model on these trials is striking in Figure 1, and it would be nice to see what ground was gained in the two-component model.

We appreciate this comment. We now have included figures showing predictive accuracy for the full model (new figures S12 and S13). The model well predicts choices following long ITIs, as well as ones following short ITIs.

*(line 220) This model well predicts choices following short and long ITIs (**Figures S12 and S13**).*

We have now also fit the kernel model to choices following long ITIs, while allowing the model to learn outcome histories over all trials. New figure S8 shows that the model can predict choices following long ITIs; but substantially fails to predict choices following short ITIs. Indeed, we found that the time constants of the model were very long (>50 trials). This supports the idea that choices following long ITIs are driven by slower learning. This is now extensively discussed in the manuscript.

*(line 150) We hypothesized that choices following long ITIs might reflect slow learning of reward history over many trials [39,34]. We first fit the same kernel model only to choices following long ITIs, by allowing the model to learn over all trials but maximizing the likelihood only from the choices following long ITIs. We found that the model could now well predict choices following long ITIs, while failing to account for choices following short ITI (**Figure S8**). Further, the time constants of the model were now very long (Reward kernel: 91 trials for WT, 59 trials for SERT-Cre mice; Choice kernel: 100 trials for WT, 143 trials for SERT-Cre mice). This supports the idea that choice following long ITIs were driven by slow learning of outcomes over many trials. We should note, however, that the difference between the choice and reward kernels becomes somewhat obscure over this timescale,*

since the reward and choice histories are strongly correlated over the long run. Thus one should take this result as inspiration, and be cautious about interpreting the precise parameter values.

We also added a figure showing how the win-stay lose switch strategy can and cannot account for choices following different durations of ITIs (new Figure S6):

(line 143) This is also evident from our additional parametric analysis showing that predictive accuracy of the win-stay lose-switch strategy dramatically decreased as ITIs lengthened (Figure S6).

I failed to understand the model-agnostic correlation analysis (Figure 2). It purports to show a correlation between two segments of data with different lengths (10 and 5 trials for rewards and choice respectively). Was this data somehow reduced to a single number for each step? The gaussian filter from Figure 1d? Please add just a bit more detail so readers can follow from Figure 2a to a pair of numbers that enter into vectors that are ultimately correlated. What kind of correlation?

We apologize for not making the description of our analysis clear. We estimated a pair of numbers (reward bias, choice bias) for each pair of sliding windows. The reward bias was computed over a 10 trial window in two conditions: trials with or without stimulation, but this was done regardless of the duration of ITIs. The choice bias was computed over a 5 trial window only on trials following long ITIs, regardless of the presence of stimulation or reward. The 10 trial reward window preceded the 5 trial choice window. By sliding these windows, we obtained many pairs of the numbers (reward bias, choice bias). We then estimated Pearson's correlation for each mouse from these numbers. This is now clarified in figure caption for Figure 2:

The correlation between the recent reward bias (window = 10 trials) and choices following long ITIs (window = 5 trials) was estimated using adjacent sliding windows. The reward bias was estimated on trials only with (top) or without (bottom) photo-stimulation, but regardless of the duration of ITIs. The choice bias was estimated only for choices following long ITIs, regardless of the presence of stimulation or reward. The greyed-out trials in this example were ignored for the assessments. The windows were shifted together one trial at a time. For each realization of the sliding windows, the reward and adjacent choice biases were estimated. However, we excluded cases in which the choice window contained no long ITI trials. By sliding these windows, we obtained many pairs of reward bias and choice bias. We then estimated Pearson's correlation from these points, separately for each mouse.

We also added a new Figure S10a which illustrates the same analysis, but only for choices following short ITIs.

I am definitely appreciate the generative model testing (Figure 3d). However, I'm a little confused in this case since there is no model of ITI generation. How did the authors run the two-component model in this case? Are there simply no short ITIs in the generated data?

We appreciate this comment. As we do not have an account of how ITIs were generated by the mouse, we simply took the ITIs from actual data when simulating the model. This is now mentioned in the manuscript:

(line 253) We used the ITIs from the real data in determining which trial was preceded by a long or a short ITI when simulating choices from the model. The ITI threshold was given by the model.

MINOR

Figure S10, the label says “all trials”, but I think it should say “<7 second ITI”.

Figure 1g, S1, the label “7<” might be easier for most to understand if it was changed to “>7”

Thank you very much. We corrected this.

Why do the authors use “partially” self-paced? It seems fully self-paced?

Trials were aborted if the mice didn't choose a side port within 100s after poking into the center port. Also, the center port was illuminated 1.1 s after the mice entered the choice port. We have now described this in the Methods section.

Reviewer #2 (Remarks to the Author):

Summary

In this manuscript the authors examine the effect of stimulating serotonin neurons on learning a self-paced probabilistic stimulus-response choice task. To do this the authors reanalyze existing data from a prior publication (Fonseca et al. Current Biol 2015) in which they examined the effect of ChR2-dependent stimulation of dorsal raphe serotonin neurons (using the Slc6a4::Cre line) on reward and waiting parameters. In this study they focus on the effect of serotonin neuron stimulation on response bias that depends on past response history. They use an error-prediction learning rate model to fit the data and estimate what type of decision strategy the animals use at each response choice. They make two major discoveries. First, they find that the animals use recent response history to guide choices when the inter-trial interval (ITI) is short (<7 s, under most circumstances) – a type of win-stay strategy, but use average response history following long ITIs (that occur sporadically). Second, they find that when they include serotonin neuron stimulation in their model it best fits the data when it influences learning rate for long ITIs. Based on these modeling data they argue that serotonin increases learning rates for reward. Understanding the role of serotonin neuromodulation in behavior is a major goal of neuroscience research with high clinical relevance. The data appear sound and are constructively imbedded in a logical experimental hypothesis and computational model and thus are an important addition to the field.

We greatly appreciate the reviewer's enthusiasm for our manuscript.

Comments:

1. The Slc6a4::Cre line they use is presumably a null allele of Slc6a4 and thus WT animals are not really the right control for these experiments. Several of the baseline, non-stimulation parameters appear to be affected by this genotype difference and the fact that the Slc6a4 mutation is likely to directly affect serotonin homeostasis, this confound is potentially problematic. The authors are aware of this point and are careful to point out where genotype may have influenced the data. However, they need to explicitly discuss the potential impact that heterozygosity of Slc6a4 could have on their findings. These animals are known to have altered serotonin tone and availability and have been widely studied as a model for the human low-expressing 5-HTT-LPR allele. Also, because there appears to be a significant difference in learning rate between WT and Slc6a4::Cre mice, there may be a ceiling effect in the WTs that confounds the data.

Sorry for not being completely clear. The reviewer might have thought that in SERT-CRe mice, the original SERT gene has been replaced by Cre, which might indeed have disturbed 5-HT homeostasis. However, the SERT-Cre line we used is a BAC transgenic line (Gong et

al., 2007), meaning that SERT-Cre gene is randomly inserted in the gene, rather than replacing the original gene. Indeed, it turns out that SERT-Cre was inserted on chromosome 18, whereas the original gene is on chromosome 11 (https://www.mmrc.org/catalog/sds.php?mmrc_id=31028).

Thus both SERT-Cre mice and WT mice have the original SERT gene, which justifies the hypothesis that the WT mice are in fact a good control. We however appreciate that there is always a possibility that the replaced gene, though not the SERT gene, had an impact on some aspects of learning and decision making in mice.

2. Although the authors are generally careful to refer to their manipulation as stimulation of serotonin neurons, on occasion they talk about stimulation of serotonin or about serotonin affecting learning rates. Given that serotonin neurons also release other neurotransmitters and these have been shown to be responsible for at least some of the phenotypes associated with stimulation of their cell bodies, the authors cannot infer that their effects are the result of changes in serotonin.

Thank you for this corrective. Indeed, we intended to talk about manipulations to serotonin neurons – and can only speculate that 5-HT itself mediates the effect. We have scoured the paper to make sure that we do not introduce any accidental solecisms along these lines.

3. The title is overly baroque and misleading. The first phrase should be eliminated as it appears to be intended to play on the presence of the S and L alleles of the serotonin transporter, but the manuscript in fact does not refer to these.

We apologize for the confusion. We did not mean the presence of the long and short alleles of the serotonin transporter, but we just meant the long and short ITIs. We changed the title accordingly.

4. Figure 1 is very small and hard to read. All the figures would benefit from being made easier for the eye.

We again apologize for this. We have now made the figures larger.

5. The methods section relies on their earlier paper for many items. Better to reiterate the critical information here (e.g. animals, methods).

Thanks very much for the suggestion. We have now included an extended summary of experimental methods in the Methods section (please see new section M1).

Reviewer #3 (Remarks to the Author):

ligaya and colleagues analyzed data from a previously published experiment (Fonseca et al., 2015) in which dorsal raphe serotonin neurons were stimulated in mice performing a foraging task. They show that serotonin activation increased learning rates following long ITIs, concluding that serotonin changes learning rates in an RL context. The previously published experimental data are beautiful, the model is conceptually exciting,

Thank you very much for the enthusiastic comments.

... but I found the conclusions vastly overstated given the data, and model selection seemed arbitrary.

We have now conducted additional analysis and clarifications to address reviewer's concerns.

1. The boundary between long and short ITIs seems arbitrary. The claim on lines 132-133 is that "choices following short ITIs and long ITIs are qualitatively different." This ultimately leads to the conclusion that "different memory mechanisms may be involved in the decisions following short and long ITIs" (lines 143-144). Is there evidence for a nonlinearity in the effects as a function of ITI that would justify the arbitrary boundary? In any case, it would be useful to see a histogram of ITIs from one session and histograms of all ITIs for each mouse.

We appreciate this comment. We now have included a histogram of ITIs from one session (new Figure S1, corresponding to the session shown in Figure 1), and histograms of all ITIs for each mouse (new Figure S2). We also included new Figures S6; S8, showing that one model can only predict either choices following short ITIs or choices following long ITIs. We also show in new figures S12 and S13 that the full model can predict choices following both short and long ITIs.

(line 108) This resulted in a wide distribution of inter-trial-intervals (ITIs). It was notable that some ITIs were substantially larger than others (Figure 1f; see also Figures S1 and S2).

(line 141) However, choices following long ITIs (ITIs > 7 s) were not well-predicted by the same model when fitting the model to all trials (Figure 1g), suggesting that choices following short ITIs and long ITIs were qualitatively different. This is also evident from our additional parametric analysis showing that predictive accuracy of the win-stay lose-switch strategy dramatically decreased as ITIs lengthened (Figure S6). This did not depend on whether long ITI trials occurred in the beginning of, or in the last part of, each experimental session (Figure S7). These results also suggest that choices following long ITIs cannot be accounted for by a short-term-memory-based win-stay lose-switch strategy.

(line 220) This model well predicts choices following short and long ITIs (Figures S12 and S13).

We agree that the ITI boundary for our model-agnostic analysis is rather arbitrary; thus we allowed the boundary to be a free parameter in our model-fitting analysis (Figure S16). We should note, though, that we introduced the hard threshold of ITI as a first approximation for capturing the non-trivial behavior that we observed. Ideally, as the reviewer rightly pointed out, the choice behavior can be characterized by subject's internal states, which may be recurrently influenced by ITIs. We now discuss this point at some length in the manuscript.

(line 359) Though as a first approximation, we assumed that a hard threshold separates ITIs for taking one choice strategy from another on following trials, we expect that this can be improved upon. For instance, which rule determines choice is presumably controlled by other variables associated with the subjects' internal states, to which we had limited access in our current study. It is also possible that both decision strategies co-exist on every trial, but their relative contributions to each ultimate decision are determined by some rules, as suggested for the integration of so-called model-based and model-free RL strategies [44]. In fact, there is evidence in macaque experiments that subjects shift to performing win-stay lose-switch if this cheap strategy offers a reasonable rate of rewards [43]. This is consistent with our finding that mice largely relied on the win-stay lose-switch, since switching behavior is known to be beneficial for this task (In fact, experimentalists often need to introduce a penalty (often in the form of a change-over delay) to deter such switching behavior [31,61]). Future studies in which the benefits and the costs of various strategies would address this issue.

2. Are the effects driven by long ITIs at the end of sessions? In this case, the effects could be interpreted as "persistence" or "task engagement," rather than learning rates, per se. The authors show that most long ITIs were at the end of sessions (Fig. S3), but do not evaluate the contribution of time within a session to the reported effects.

We appreciate this comment. We tested if choices following long ITIs at the end of the sessions were different from choices following long ITIs in the beginning of the sessions. We however found null results (new Figure S7). This is now discussed in the manuscript.

(line 143) This is also evident from our additional parametric analysis showing that predictive accuracy of the win-stay lose-switch strategy dramatically decreased as ITIs lengthened (Figure S6). This did not depend on whether long ITI trials were in the beginning of, or in the last part of, each experimental session (Figure S7).

Given that the ITIs were self-generated, it is difficult to disentangle forgetting (presumably a nondecreasing function of time) from "motivation" (a nonincreasing function of time, as the animal gets less thirsty). A new experiment, with experimenter-generated ITIs, could potentially resolve this.

We tested a model that learns and forgets reinforcement history in real time, instead of on a trial-by-trial basis. However, this model performed significantly worse than the conventional trial-by-trial model.

*(line 226) One might wonder if the behavior could be better accounted for by a model specifying forgetting as a function of elapsed time, including the ITIs. To test this, we constructed a model that learns and forgets outcome history according to wall-clock time (measured in seconds) rather than according to the number of trials. For this, we simply adapted the previously validated two-kernel model that integrates choice and reward history over trials [42,23] such that the influence of historical events is determined by how many seconds ago they happened, using the factual timing of the experiments. Our model comparison analysis using WT mice, however, substantially favored the account of trial-based model **Figure 3a** ($\Delta\text{iBIC} = 218$). Introducing two time constants to the reward integration kernel did not change this conclusion.*

The new experiment suggested by the reviewer is interesting. However, we suggest that there would remain a circularity – since motivation can be influenced by ITIs just as ITIs can be influenced by motivation. We would therefore prefer to leave this thorny problem to future studies.

3. Did the RL model with slow learning still fit the behavior following short ITIs? It seems very strong to conclude different memory mechanisms due to quantitatively different fits (BIC score differences). Indeed, a more parsimonious explanation would be simply that the new model is better at describing behavior than the old one (which, by itself, would be interesting). If the authors believe there are two separate memory mechanisms (issues above notwithstanding), how would it work for the brain to "choose" one over the other in real time? Is the claim that $T_{\text{Threshold}}$ is implemented neurally? Why would serotonin affect one but not the other? Why not parameterize ITI in the model, as opposed to using a threshold value?

We now have fit the kernel model to choices following long ITIs, while allowing the model to learn outcome histories over all trials. New figure S8 shows that the model can predict choices following long ITIs; but it substantially fails to predict choices following short ITIs. Indeed, we found that the time constants of the model were very long (>50 trials). This

supports the idea that choices following long ITIs are driven by slower learning. This is now extensively discussed in the manuscript.

(line 150) We hypothesized that choices following long ITIs might reflect slow learning of reward history over many trials [39,34]. We first fit the same kernel model only to choices following long ITIs, by allowing the model to learn over all trials but maximizing the likelihood only from the choices following long ITIs. We found that the model could now well predict choices following long ITIs, while failing to account for choices following short ITI (Figure S8). Further, the time constants of the model were now very long (Reward kernel: 91 trials for WT, 59 trials for SERT-Cre mice; Choice kernel: 100 trials for WT, 143 trials for SERT-Cre mice). This supports the idea that choice following long ITIs were driven by slow learning of outcomes over many trials. We should note, however, that the difference between the choice and reward kernels becomes somewhat obscure over this timescale, since the reward and choice histories are strongly correlated over the long run. Thus one should take this result as inspiration, and be cautious about interpreting the precise parameter values.

One reason that we mentioned different memory mechanism is that fast system was described by RL including a forgetting model (kernel model), while the slow system was better described by RL without forgetting model. The former indicates working memory, while the latter may not. However, we definitely agree that more studies would be necessary to pin down the memory mechanisms that are involved.

We also do not believe that animals have a fixed, hard threshold for the ITI to arbitrate two decision modes, as it depends on animals' internal states and the threshold could also be more gradual and stochastic. We should also point out that our model could not capture the effect of much slower learning that we observed in data (figures S18, S19, S20). Thus in reality, learning may be taking place over multiple (more than two) timeconstants in parallel, of which the shortest is well-captured by the win-stay lose-switch strategy. In turn, animals' choice is determined by a weighted average of what is learned over multiple time constants, as recently suggested in a similar experimental paradigm in macaque [Iigaya, Ahmadian, Sugrue, Corrado, Newsome, Fusi, bioRxiv 141309 (2017)], where subjects used reward histories of 1 trial to thousands of trials at the same time. It could be that the relative weights are strongly influenced by the duration of ITIs. Thus making the hard threshold be a free parameter, as we did, is a very crude first approximation of what may actually be happening. We however believe that our study is a first, but non-trivial, step toward more refined understanding of how ITIs, and indeed animal's internal states between trials, can modulate the impacts of learnings over distinctive timescales on choice.

We covered this in the Discussion:

(line 359) Though as a first approximation, we assumed that a hard threshold separates ITIs for taking one choice strategy from another on following trials, we expect that this can be improved upon. For instance, which rule determines choice is presumably controlled by other variables associated with the subjects' internal states, to which we had limited access in our current study. It is also possible that both decision strategies co-exist on every trial, but their relative contributions to each ultimate decision are determined by some rules, as suggested for the integration of so-called model-based and model-free RL strategies [44]. In fact, there is evidence in macaque experiments that subjects shift to performing win-stay lose-switch if this cheap strategy offers a reasonable rate of rewards [43]. This is consistent with our finding that mice largely relied on the win-stay lose-switch, since switching behavior is known to be beneficial for this task (In fact, experimentalists often need to introduce penalty for such switching behavior, which is often referred to as the change-over-delay

[31,61]). *Future studies in which the benefits and the costs of various strategies would address this issue.*

REVIEWERS' COMMENTS:

Reviewer #1 (Remarks to the Author):

The authors have done a good job responding to my comments. I appreciate their efforts and feel the additional work clarifies several important points. I have no further comments, and recommend the paper for publication.

Reviewer #2 (Remarks to the Author):

The authors have satisfactorily addressed my concerns.

Reviewer #3 (Remarks to the Author):

The authors have addressed all my previous comments.

REVIEWERS' COMMENTS:

Reviewer #1 (Remarks to the Author):

The authors have done a good job responding to my comments. I appreciate their efforts and feel the additional work clarifies several important points. I have no further comments, and recommend the paper for publication.

Reviewer #2 (Remarks to the Author):

The authors have satisfactorily addressed my concerns.

Reviewer #3 (Remarks to the Author):

The authors have addressed all my previous comments.

Thank you very much for the very positive reviews.